# Ageing-Related Changes to H3K4me3, H3K27ac, and H3K27me3 in Purified Mouse Neurons

**DOI:** 10.3390/cells13161393

**Published:** 2024-08-21

**Authors:** Brandon Signal, Andrew J. Phipps, Katherine A. Giles, Shannon N. Huskins, Timothy R. Mercer, Mark D. Robinson, Adele Woodhouse, Phillippa C. Taberlay

**Affiliations:** 1Menzies Institute for Medical Research, University of Tasmania, 17 Liverpool Street, Hobart, TAS 7000, Australia; brandon.signal@utas.edu.au (B.S.); kgiles@cmri.org.au (K.A.G.); shannon.ray@utas.edu.au (S.N.H.); 2Wicking Dementia Research and Education Centre, College of Health and Medicine, University of Tasmania, 17 Liverpool Street, Hobart, TAS 7000, Australia; andrew.phipps@utas.edu.au; 3Children’s Medical Research Institute, University of Sydney, 214 Hawkesbury Road, Westmead, NSW 2145, Australia; 4Australian Institute for Bioengineering and Nanotechnology, Corner College and Cooper Roads, Brisbane, QLD 4072, Australia; t.mercer@uq.edu.au; 5SIB Swiss Institute of Bioinformatics, University of Zurich, Winterthurerstrasse 190, CH-8057 Zurich, Switzerland; mark.robinson@mls.uzh.ch

**Keywords:** neurons, aging, epigenetics, histones, ChIP-Seq

## Abstract

Neurons are central to lifelong learning and memory, but ageing disrupts their morphology and function, leading to cognitive decline. Although epigenetic mechanisms are known to play crucial roles in learning and memory, neuron-specific genome-wide epigenetic maps into old age remain scarce, often being limited to whole-brain homogenates and confounded by glial cells. Here, we mapped H3K4me3, H3K27ac, and H3K27me3 in mouse neurons across their lifespan. This revealed stable H3K4me3 and global losses of H3K27ac and H3K27me3 into old age. We observed patterns of synaptic function gene deactivation, regulated through the loss of the active mark H3K27ac, but not H3K4me3. Alongside this, embryonic development loci lost repressive H3K27me3 in old age. This suggests a loss of a highly refined neuronal cellular identity linked to global chromatin reconfiguration. Collectively, these findings indicate a key role for epigenetic regulation in neurons that is inextricably linked with ageing.

## 1. Introduction

Ageing is characterised by the gradual loss of physiological integrity, resulting in the impaired function of bodily systems. In the central nervous system, ageing is associated with variable cognitive decline in healthy individuals. The activity of neurons in the brain underlies cognitive function, and like many bodily systems, the ageing of the central nervous system is linked to functional and structural vicissitudes in the neurons [1,2]. Neurons are one of the few cell types in the body that can subsist for the entire lifespan, with functional plasticity retained to enable lifelong learning and memory [3,4,5], although low-level adult neurogenesis has been observed in the cortex of mice [6,7].

The epigenome is at the interface of genes and the environment, enabling the dynamic regulation of gene expression in each cell. The epigenome includes DNA methylation and histone modifications, as well as the physical organisation of DNA inside the nucleus, which dynamically regulates gene expression [8]. Histone modifications are associated with an altered chromatin architecture, with certain modifications commonly found in particular active or repressed genomic elements. The histone modification H3K4me3 is a well-studied marker of active gene promoters [9,10]. H3K27ac, H3K4me1 [11], H3K4ac [12], and H3K9ac [13] similarly mark active gene promoters as well as enhancers. H3K36me3 marks active gene bodies [14], as well as having roles in splicing and the DNA damage response [15]. The broad histone modifications H3K27me3 and H3K9me3 typically mark repressed regions and are associated with facultative and constitutive heterochromatin, respectively [16]. While these associations are known, the combination of epigenetic elements determines whether a genomic region is active or repressed.

Several studies have established that age-associated epigenetic alterations occur in the brain [17,18,19,20] and are linked with age-related cognitive decline [3,4,5]. Notably, the loss of histone acetylation is correlated with age-associated memory deficits [21], and the inhibition of histone de-acetylation pathways can reverse age-associated cognitive deficits [22,23,24]. However, the brain comprises a complex mixture of cell types, of which ~50% are neurons. Each cell type has a unique epigenetic signature and exhibits distinct changes to the epigenome with ageing, meaning that these cell types should be studied individually to better understand epigenetic changes in the brain [17,19,25,26,27,28]. The activation of inflammatory pathways into old age identified in mouse whole-brain samples indicates that the epigenetic signal from glial cells, which have roles in neuro-immune systems, may mask the detection of epigenetic changes in neurons [20,29]. This is supported by non-neuronal H3K4me3 and H3K27ac profiles clustering more closely with homogenate-derived profiles and neuronal-derived profiles being more distinct [30]. Moreover, single-cell RNA sequencing has identified patterns of gene expression that are regulated in opposite directions with advanced ageing in different mouse brain cell types [31]. Few genome-wide epigenetic maps of purified neurons exist from adulthood into old age, particularly in mice [17,19,25,32], although these studies have shown that neurons exhibit dynamic epigenome remodelling from prenatal development to adulthood [32].

There is notably a lack of the commonly studied H3 histone marks K4me3, K27ac, and K27me3 in mouse neurons that have been purified. To address one aspect of this lack of data, we produced maps of the H3K4me3, H3K27ac, and H3K27me3 dynamics in purified neurons across the lifespan in C57/BL6 male mice. The locations of these three marks enable the identification of active promoters, facultatively repressed regions, and putative enhancers—which are key facets of dynamic genome regulation. While they are stand-alone datasets, these maps provide evidence of highly consistent H3K4me3 throughout advanced ageing, and a striking loss of H3K27ac from synaptic signalling genes and H3K27me3 from developmental genes. Collectively, these findings shed light on the pivotal role of epigenetic reconfiguration in neuronal ageing.

## 2. Materials and Methods

### 2.1. Mouse Tissue Processing

All animal procedures were undertaken with ethical approval from the Animal Ethics Committee of the University of Tasmania (A12780/A15120) and all experiments abided by the Australian Code of Practice for the Care and Use of Animals for Scientific Purposes. Male mice for ChIP-Seq were aged 3 months (young adult), 12 months (middle-aged adult), or 24 months (old-age adult), with *n* = 5 mice per time point. The animals were housed under standard conditions (12 hr day/night cycle, housing temperature of 20 °C, and ad libitum access to food). The mice were sacrificed with an intraperitoneal injection of Lethobarb (110 mg/kg) and cardiac perfusion with 0.01 M PBS. The forebrain was then rapidly dissected and snap-frozen in liquid nitrogen.

### 2.2. Isolation of Nuclei from Mouse Forebrain

Fresh-frozen mouse forebrains were sectioned along the midline. Half of each forebrain was immersed in 4.8 mL of nuclei extraction buffer (NEB [sucrose, 0.32 M; CaCl_2_, 5 mM; Mg(Ac).4H_2_O, 3 mM; EDTA, 0.1 mM; Tris-HCl, pH of 8.0, 10 mM; 1X protease inhibitor cocktail, PMSF, 0.1 mM; Triton-X-100, 0.1%], Ipswich, MA, USA) on ice and homogenised with a dounce tissue homogeniser (#357544, Wheaton, Millville, NJ, USA). The homogenised tissue was filtered through four layers of cheesecloth (#9338918004577, Ogilvies Designs, Belmont, Western Australia, Australia), followed by 70 μm and 40 μm cell strainers (#352350 and #352340, Thermo Fisher Scientific, Waltham, MA, USA) on ice. The nuclei were pelleted by centrifugation and resuspended in 500 μL of 0.01 M PBS. The nuclei (90%) were incubated with anti-NeuN antibodies (1:1000, #MAB377, Merk Millipore, Billerica, MA, USA) and goat-anti-mouse AlexaFluor 647 (1:2000, #A31571, Invitrogen/Life Technologies, Carlsbad, CA, USA) in a blocking solution (0.5% BSA, #B4287 25 G, Sigma-Aldrich, St Louis, Missouri, USA, and 10% normal goat serum, #G9023-10, Sigma81 Aldrich, St Louis, MI, USA, in 0.01 M PBS) for 40 min at 4 °C on a slow rotator in the dark. The remaining 10% of the nuclei were used as the secondary-only control and were incubated with goat-anti-mouse AlexaFluor 647 (1:2000) in the blocking solution for 40 min at 4 °C on a slow rotator in the dark. DAPI (#D3571, Life Technologies, Carlsbad, CA, USA) was added to the secondary-only control for the final 5 min of the incubation. Following the antibody incubations, the nuclei were collected by centrifugation and then resuspended in ice-cold PBS.

### 2.3. Fluorescence-Activated Nuclei Sorting (FANS) for Neuronal Nuclei

FANS was performed on a BD Biosciences FACS Aria III (Becton Dickinson Biosciences, Franklin Lakes, NJ, USA). The secondary-antibody-only negative controls established gating for each sample, and NeuN-labelled nuclei were then sorted and collected in tubes pre-coated with foetal calf serum (FCS; #SH30084.02, Hyclone, GE Healthcare Life Sciences, South Logan, UT, USA; Appendix A). Purity checks were performed, and the samples were validated at >97% purity (Appendix A).

### 2.4. Chromatin Immunoprecipitation (ChIP)

The NeuN+ nuclei were fixed in 1% methanol-free formaldehyde (#28906, Pierce, Thermo Fisher Scientific, Waltham, MA, USA) for 15 min at room temperature, and then quenched with 0.125 M glycine (#G8898, Sigma-Aldrich, MI, USA) for 5 min, prior to freezing at −80 °C. The neuronal nuclei were then collected by centrifugation at 52,000× *g* at 4 °C for 25 min (Sorvall WX Ultra 90 ultracentrifuge; TH-641 swinging bucket rotor (#54295, Thermo Fisher Scientific, Waltham, MA, USA)), resuspended in SDS lysis buffer (EDTA, 10 mM; SDS, 1%; Tris-HCl, pH of 8.1, 50 mM), and transferred to 1.5 mL Bioruptor+ TPX microtubes (#C30010010-300, Diagenode, Seraing, Belgium). The samples were sonicated on a Bioruptor Plus next-gen ultrasonicator (Diagenode, Seraing, Belgium) for 50 cycles of 30 s on/30 s off, on ice. The chromatin fragment size was confirmed to be 200–500 bp on the Agilent 4200 tapestation with HS D1000 tape and reagents (#5067-5584, Agilent Technologies, Santa Clara, CA, USA). Chromatin immunoprecipitation was performed as previously described [33,34,35] with minor modifications. Briefly, 5 × 10^5^ nuclei were used per immunoprecipitation. All the samples (*n* = 20) were processed in parallel to eliminate batch effects. The samples were incubated with the primary antibodies anti-H3K4me3 (2 μg; #39160; Active Motif, Carlsbad, CA, USA), anti-H3K27ac (2 μg; #39134; Active Motif, Carlsbad, CA, USA), or anti-H3K27me3 (07-449; Millipore, Burlington, MA, USA) overnight at 4 °C with rotation. All the antibodies were previously validated for their specificity and use in ChIP assays.

### 2.5. ChIP-Seq Library Preparation and Next-Generation Sequencing

Sequencing libraries were prepared using the Nugen Ovation Ultralow V2 (#0347 V2 1-96/0344 V2 1-16, Redwood City, CA, USA) library preparation kits as per the manufacturer’s protocol. Library quality control and normalisation were carried out by the Australian Genomics Research Facility with the Agilent 4200 tapestation system and KAPA qPCR quantification. The samples were then pooled and distributed equally across all sequencing lanes. The samples were sequenced on an Illumina Hi-Seq 2500 Next-Generation Sequencer (Illumina, San Diego, CA, USA) using 50 bp single-end sequencing.

### 2.6. ChIP-Seq Alignment

The raw reads were subjected to quality control with FastQC (v0.11.7) [36] and MultiQC (v1.12) [37]. The adapters were trimmed using the Trim Galore wrapper (v0.4.3) for Cutadapt (v1.15) [38]. The reads were aligned to the mm10 genome using Bowtie2 (v2.3.5.1) [39], and alignments with an MAPQ lower than 30 were discarded using the samtools view (v1.16.1) [40]. For H3K27ac and H3K4me3, MACS2 (v2.2.7.1) [41] was used to call peaks against a total input control sample, with an initial minimum q-value of 0.001. For H3K27me3, SICER2 (v1.0.3) [42] was used to call peaks against a total input control sample, with a minimum FDR of 0.05, a window size of 500, and a fragment size of 300. The peaks were removed if they overlapped mm10 blacklisted regions [43]. ChIPQC (v1.30.0) [44] was used to compare the alignment and peak metrics. Bam files were converted to bigwigs using deeptools bamCoverage (v3.5.1) [45] with window widths of 50 nt (for plotting) and 1000 nt (for read alignment calculations and PCA), and normalised using RPKM.

### 2.7. Annotation Datasets

A general genomic feature dataset was generated using annotatr (v1.28.0) [46], covering the FANTOM enhancers [47] and all mm10_gene features except intron-centric annotations. An expanded enhancer annotation was derived from the ENCODE postnatal day 0 midbrain, forebrain, and hindbrain 18-state ChromHMM annotations [48], keeping the regions marked as Enh, EnhG, EnhLo, EnhPois, or EnhPr. The non-disjoint FANTOM enhancer, ENCODE enhancer, extended promoters (1−5 kb from TSS), and intergenic regions were exported in bed format to calculate the read alignment depth in these regions using Mosdepth (v0.2.6) [49]. The enhancer annotations were merged with the annotatr dataset, and “introns” and “intergenic” regions were re-annotated by disjoining these regions with any enhancers. Superenhancers for the mouse brain (E14.5 brain, cerebellum, and cortex) were obtained from dbSUPER [50] and converted from mm9 to mm10 coordinates using the mm9-to-mm10 liftover chain file from UCSC and rtracklayer’s (v1.62.0) liftOver function [51].

### 2.8. Cell-Identity Gene Expression and ChIP-Seq Coverage

Genes identified by Ximerakis et al. [31] were used as markers of cellular identity for five major cell types. We note that, as these were identified using differential expression by comparing one cell type to all others, these genes are not always exclusively expressed in one cell type and have some overlap in expression with other similar cell types. Moreover, as this comparison combined both younger (2–3-month-old) and older (21–22-month-old) mice, these genes should be expressed in both younger and older cells. We required a minimum average log2 fold change of 1 compared to other cell types to define our set of cellular identity genes. For H3K4me3 or H3K27ac to be considered as covered at the promoter of a gene, a minimum RPKM of 2 in a 1 kb window overlapping the region +/−2 kb from the transcription start site of any protein-coding isoforms of the parent gene with a Gencode vM23 transcript support level of 1 or 2 was required [52]. For H3K27me3, the median RPKM values in the 1 kb windows over the entire gene body were calculated, and the median of these values was used for plotting.

### 2.9. Sample Quality Control and Filtering

Samples were included based on reasonable quality control filters (Appendix A). For H3K4me3, an active promoter mark, we required the Mosdepth/annotatr read depth ratio between the promoters and intergenic regions to be greater than 1.5, at least 95% of the mature neuron marker gene promoters to have an average RPKM of 2 across a 1 kb window, at least 10,000 peaks (q < 0.001), and 100,000 high-quality (MAPQ ≥ 30) aligned reads. For H3K27ac, an active mark that is typically found at enhancers, we required the Mosdepth/annotatr read depth ratio between the enhancers and intergenic regions to be greater than 1.1, a fragment length of at least 150, at least 1000 peaks (q < 0.001), and 200,000 high-quality (MAPQ ≥ 30) aligned reads. For H3K27me3, a repressive mark, we required at least 500 peaks and 500,000 high-quality (MAPQ ≥ 30) aligned reads, as well as an average pairwise overlap in the peaks of at least 10% in a quarter of the other samples. The investigation of the cellular identity gene signal in H3K27ac and H3K4me3 is shown in Appendix A, and it showed that the majority of the discarded samples did not have a sufficient signal representative of any brain cell subtype used. Furthermore, the correlation of the samples in the first component of the PCAs of 10 kb windows for each ChIP mark (Appendix A) indicated that these low-quality samples were typically of a very poor quality across all experiments and likely did not contain sufficient high-quality material to perform ChIP-Seq representative of the sample. The histone modification profiles for each sample passing QC were generated against the merged total input controls, using deeptools bamCompare, computeMatrix, and plotProfile with a bin width of 10 [45]. The histone modification profiles showed similar patterns in intensities when profiled across all genes (Appendix A).

### 2.10. Genomic Feature Overlap Calculations

A custom function was used to determine which genomic feature a peak or region was most proximal to or overlapped. The ChIPpeakAnno (v3.36.1) [53] function annotatePeakInBatch was used to find such regions. As this frequently yielded multiple hits, we devised a consistent ranking method to determine which feature had the closest relationship to the query region. The overlap width was calculated as the width of intersecting regions. The distance to the nearest feature was calculated from the “shortestDistance” variable, with any overlapping regions being reset to 0. Relationships were ordered as a factor in the following order: “includeFeature”, “inside”, “overlapStart”, “overlapEnd”, “upstream”, and “downstream”. Hits were ranked first based on the overlap width, then by the distance to the nearest feature, by the intergenic status (where all other overlap types were prioritised over intergenic overlaps), and finally, by relationship (in the order given above). Over- and underrepresentations of overlap types were calculated by generating a random set of 10,000 peaks with the same widths that fell within the minimum and maximum chromosomal boundaries of peaks. These “random” peaks were annotated with the same method as described above, and a hypergeometric test comparing these to the original peak set was applied to each category.

### 2.11. Gene Ontology Enrichment

The gene ontology enrichment was performed using ChIP-Enrich (v2.26.0) [54] and the mm10 annotation.

### 2.12. Motif Analysis

The regions were converted to fasta sequences using Biostrings (v2.70.3) [55] and exported as fasta sequences. The motifs were found using the findMotifs.pl script from HOMER (v4.11) [56], with the vertebrate set of known motifs. For the novel H3K27ac peak results presented, the de novo motif results were used. For the differential H3K27me3 peak results presented, the known motif results were used. The motifs were filtered out of the results if they contained more than 40% of the same 1-, 2-, or 3-mer repeat.

### 2.13. Additional Datasets

ENCODE H3K27ac pseudoreplicated peak bed files were downloaded for all adult mouse brain samples [57]. Strand-specific ENCODE RNA-Seq wiggle tracks for mice were downloaded for all adult left cerebral cortex samples and postnatal day 0 forebrain, midbrain, and hindbrain samples [48]. A list of ENCODE file IDs is given in Appendix A. The snATAC peak data were obtained from [58] using their Appendix A, which contained a list of cCREs and their cell-type assignment, respectively. A list of the cell-type abbreviations used within this manuscript and detailed descriptions of the cell types are given in Appendix A.

### 2.14. Differential Peak Analyses

Differential peak analyses were performed using csaw (v1.26.0) [59]. The counts were read in using windowCounts with a fragment length of 250, a window width of either 200 (H3K27ac, H3K4me3) or 1000 (H3K27me3), and a minimum quality of 30, and windows overlapping the mm10 blacklist were removed. These counts were filtered using the global enrichment approach, using a window width of 10,000 to calculate background enrichment, and the bins were retained if they had a fold change of 3 over the background. The counts were normalised using non-linear normalisation. Differential histone modifications were tested using the glmQLFTest [60] function; the windows were merged using mergeResults, with a window width of 1 kb; and the window-level *p*-values were computed using combineTests. For comparisons of the ChIP signal, the normalised counts were extracted for each window and averaged across the differential window, and the median value for each age was used to calculate the fold change difference (old/young). For H3K27ac, as some regions did not pass the csaw filters when applied to all three ages simultaneously, a window coverage of at least 75% was required.

### 2.15. ChIP-Seq Profile Plotting

Coverage bigwig files with a width of 50 nt were used for plotting coordinate-specific ChIP-Seq profiles. The bigwigs were imported into R using rtracklayer’s import.bw [51], and a rolling mean of RPKM coverage was calculated across 250 nt windows. The transcripts were plotting using the Gencode vM23 annotation [52], requiring a transcript support level of 1, and the exon locations for all the transcript isoforms for a gene meeting this requirement were merged into a single track. Enhancers were plotted as a single track using the ENCODE brain enhancer annotations described previously. Superenhancers were plotted as a single track using the DBSuper superenhancer annotations described previously [50].

## 3. Results

### 3.1. Generation of Neuronal-Specific Epigenome Maps during Mouse Ageing

To address the current gap in our understanding of neuronal-specific epigenetics during the adult lifespan, we utilised fluorescence-activated nuclei sorting (FANS) to purify NeuN^+^ (RNA-binding protein, fox-1 homolog; Rbfox3 positive) neuronal nuclei (Appendix A). We applied this technique to the forebrains of C57/BL6 mice with an age of 3 months (young adult), 12 months (middle age), and 24 months (old age; Figure 1A,B). Post-purification, the chromatin of the neuronal nuclei was immediately cross-linked, and then subjected to chromatin immunoprecipitation and sequencing (ChIP-Seq; Figure 1C,D) to profile the dynamics of H3K4me3, H3K27ac, and H3K27me3 across the adult lifespan. These three marks were specifically chosen to enable the profiling of active promoters and enhancers as well as facultatively repressed regions with the limited cellular material available post-neuron purification, giving a more expansive view of how gene regulation through multiple elements changes with ageing.

To confirm that our FANS neuronal nuclei purification was enriched for a neuronal-specific epigenetic signature, we investigated the H3K4me3 and H3K27ac signals at gene promoters. Following stringent quality control processing and filtering of the ChIP-Seq samples (see Methods) which resulted in between two and four biological replicates per age and histone modification, we examined the H3K4me3 and H3K27ac signals (>2 RPKM) at cellular identity genes (unique molecular identifiers, 206–404 genes per cell type) from Ximerakis et al. [31]. This demonstrated the enrichment of these active marks at neuronal cell promoters over all other brain cell types (Figure 1E). Specifically, we found robust H3K4me3 and H3K27ac signals at the promoters for the neuronal-specific genes *Syt1* and *Syp* [61,62,63], and no signals at the glial-specific genes *Gfap* and *Aif1* (Appendix A) [64].

Furthermore, the H3K27me3 ChIP signal over the bodies of these same genes showed the expected inverse pattern, with a lower signal at neuronal genes (Figure 1E). Following peak calling, the closest annotated genomic features were aligned with known deposition locations (Appendix A). These features were typically stable across age; however, we noted a minor loss of otherwise unannotated intergenic regions, a gain of promoter H3K27ac (chi-squared *p* < 2.2 × 10^−16^), and a loss of annotated enhancers with a concurrent gain of unannotated intergenic regions of H3K27me3 with age (chi-squared *p* < 2.2 × 10^−16^). A GO enrichment analysis using software designed for ChIP peaks revealed that the H3K27ac peaks were enriched for intracellular transport and synaptic signalling (Appendix A). The H3K4me3 peaks were enriched for intracellular transport and RNA processing terms across all age groups (Appendix A). The enrichment of intracellular transport terms in the two active marks was aligned with the fundamental role this has in neuronal function due to the complex morphology of neurons. For the repressive mark, H3K27me3, the observed enrichment of developmental terms, and later sensory perception was reflective of the repression of these genes and regulatory elements and was mirrored by a concurrent depletion in these same terms in the H3K4me3 peak sets, indicating the coordinated epigenetic control of these processes (Appendix A). Therefore, our data show the expected pattern of active mark enrichment at neuronal marker genes and for all marks with the expected biological processes, and they support our preparation of purified neurons. A principal component analysis (PCA) of the global signal for all the histone marks profiled showed a clear separation between young-adult (3 months) and middle-aged/old-age (12/24 months) samples, with the middle-aged and old-age samples often clustered together (Figure 1F). Lastly, there was considerable overlap between H3K27ac and H3K4me3, with 59% of the H3K4me3 peaks co-occurring with H3K27ac (Appendix A), which is indicative of these modifications co-marking active regions [65]. In addition, there was a small amount of overlap between H3K4me3 and H3K27me3 (0.9–2.5% peak co-occurrence; Appendix A), which could mark bivalent chromatin [66], and minimal overlap between H3K27ac and H3K27me3 (0.2–0.6% peak co-occurrence; Appendix A), which was expected, as these modifications should be exclusionary.

### 3.2. Purified Neurons Enable Detection of an Expanded Repertoire of H3K27ac-Marked Neuronal Enhancers

Following confirmation that the neuronal H3K27ac peaks typically co-occurred with brain-derived ENCODE enhancers and FANTOM enhancers, we observed that a number occurred in otherwise unannotated intergenic (up to 13%) or intronic (up to 15%) regions (Appendix A). To elucidate which of these represent novel neuronal H3K27ac-marked regions, we first merged the peaks across all ages into a single meta-peak set, keeping those with evidence from at least two samples (Appendix A). We then compared these peaks to multiple datasets, including whole-brain ChromHMM features and H3K27ac as well as single-cell ATAC, confirming the neuronal enrichment and brain region specificity and allowing for the identification of regions with minimal previous evidence for enhancer activity (Appendix A). From this, we identified a set of novel peaks that were not located nearby a ChromHMM enhancer or overlapping with an ENCODE H3K27ac peak or ATAC-accessible regions. These novel H3K27ac peaks, comprising 1685 (6%) of our meta-peak set, were most commonly found in introns, typically did not co-occur with paired neuronal H3K4me3 (<20%), and occurred in and nearby genes relating to synaptic and ion transport functions (Figure 2A,B and Appendix A). A HOMER motif analysis showed enrichment for several neurogenesis and neurodevelopmental transcription factor-binding sites [67,68,69,70,71,72,73], as well as sites linked to oxidative stress and neurodegeneration [74,75,76], commonly marking the intronic regions of known synaptic genes, including those with links to neurological disease (Figure 2C,D) [77]. As intronic enhancers typically have tissue-specific activity [78] and given these enrichments, the majority of these novel H3K27ac peaks likely represent previously unannotated enhancers, which were formerly masked by whole-brain-derived ChIP-Seq [29] and are important in neuronal function.

### 3.3. Differential Histone Modification Reveals a Redistribution of H3K27ac from Intronic Enhancers to Promoters with Neuronal Age, and a Loss of Repressive H3K27me3 from Developmental Genes

We next sought to determine which genome regions exhibited variations in histone modification. In our analysis of H3K27ac, we observed thousands of regions with reduced levels between young adults and middle-aged (2221) or old-age (2994) adults, and a single region with a gain in the H3K27ac signal between middle-aged and old-age samples (Figure 3A and Appendix A). This single region gained between middle-aged and old-age samples occurred in the first intron of the zinc finger gene *Zswim6*, which has been previously linked to neurodevelopmental disorders and schizophrenia (Figure 3B and Appendix A) [79,80], and although statistically significant, we noted that the gain appeared to only occur to a noticeable degree in one of the two 24-month samples. As this differential peak analysis was only able to detect this single change between 12 and 24 months, this suggests that the majority of H3K27ac configuration is stable or undergoes smaller changes from middle-aged to old-age mice. Regions showing a lower H3K27ac with age typically overlapped with annotated enhancer regions (Figure 3C); the majority (66%) did not overlap with H3K4me3 peaks; and, as expected, the vast majority (99.7%) did not overlap with H3K27me3 peaks (Appendix A), which is consistent with mutual exclusivity. Those that did overlap with H3K27me3 peaks, while rare, were typically encompassed by a much wider H3K27me3 peak (median of 16x wider than the H3K27ac region) and, therefore, are not likely to represent truly paradoxical histones with both modifications. The H3K27ac regions lost from young-adult to old-age neurons were enriched for synaptic signalling and ion transport (Figure 3D). These changes were consistent with single-cell mouse neuronal gene expression data at similar ages (Appendix A) [31]. Between the young-adult and middle-aged stages, the H3K27ac regions lost were enriched for GO terms relating to central nervous system neuron development and chromatin disassembly (Figure 3D). This suggests an intertwining of global H3K27ac reconfiguration with the regulation of other epigenetic changes during neuronal ageing. There was some overlap in the peaks lost between 3 and 12 months and between 3 and 24 months (Figure 3E), and the quantification of the ChIP signal in all differential regions across all ages suggested that the peaks lost between 3 and 24 months follow a pattern of consistent loss, and those lost between 3 and 12 months follow a pattern of sustained loss (Appendix A).

Surprisingly, a differential binding analysis of H3K4me3 found only a single region with significantly different enrichment between any two ages, suggesting that the H3K4me3 deposition patterns in neurons are highly consistent from young adults through to old age (Figure 4A). The single loss between young adults and old age occurred within an intronic region of *Igf2bp1* nearby an ENCODE midbrain enhancer (Figure 4B). Although H3K4me3 is typically an active promoter mark, this falls outside the annotated *Igf2bp1* promoter region and is not marked by high levels of H3K27ac or H3K27me3 in any age (Appendix A). Igf2bp1 is an RNA binding protein that binds to *β*-actin [81]. It is required for growth cone migration in neurons and is involved in dendritic branching, synapse formation, and long-term potentiation [82,83,84,85,86,87], making this gene and region functionally relevant to neuronal biology. Given this profile, we sought to determine if there was any evidence for RNA transcription in this region, by finding ENCODE postnatal cortex samples that display low levels of an RNA signal, albeit from the opposite strand (Appendix A). This may be reflective of a non-coding antisense transcript involved in *Igf2bp1* regulation, given both the prevalence of non-coding transcription in the brain and the propensity for intronic antisense transcripts to regulate their parent genes [88,89,90,91], which warrants further characterisation given the distinct lack of other H3K4me3 changes during advanced neuronal ageing.

Finally, we found 588 regions with a lowered H3K27me3 signal intensity (and a lack of any regions gained) between young-adult and old-age neurons, which occurred predominantly in gene promoters (Figure 5A). Such promoter regions (−5 kb to +1 kb from TSS) were overrepresented by CpG island promoters. These regions were frequently observed at homeobox (HOX) genes, and were enriched for embryonic development and transcription factor binding (Figure 5B,C and Appendix A). In addition to regions encompassing several developmental transcription factor genes, a motif analysis of promoter regions showed the enrichment of their resulting product’s binding motifs, including Wt1 [92], Hand2 [93], and Foxd3 [94] (Appendix A). While such developmental-related enrichments were the most prominent, there was also enrichment for both neuronal-related GO terms (e.g., GO:0007417 central nervous system development, FDR = 8 × 10^−58^, and GO:0048663 neuron fate commitment, FDR = 3 × 10^−39^) and motifs (e.g., Egr1 [95], Klf9 [96], and Ebf2 [97]), suggesting that these neuronal pathways may be dysregulated with age (Appendix A). While the csaw analysis was not constrained by pre-annotated peak regions, 98.6% of the peak changes overlapped with an H3K27me3 peak. A single region with a decreased signal was found in the comparison between middle-aged and old-age neurons, which occurred within the intron of an unannotated gene (*Gm2420*; FDR = 0.006) and did not overlap with the H3K27me3 peak set. This suggests weak evidence of change. No regions showed statistically significant changes in H3K27me3 between 3 and 12 months. The regions that lost the H3K27me3 signal between 3 and 24 months showed a similar direction of loss between 3 and 12 months, although the changes were not statistically significant between these ages (Appendix A).

Unlike H3K27ac, the regions lost between young-adult and old-age neurons were more frequently lost from the superenhancers annotated in the E14.5 brain, and not the cortex (Figure 5D, Appendix A). These changes were typically not reflective of a significant change in gene expression at similar ages (Appendix A) [31]. In concert with the observed GO enrichment for the lost regions, this suggests that, as is well established, H3K27me3 has a role in suppressing embryonic developmental genes in young-adult neurons, but that during ageing, the regulation of these regions with H3K27me3 is relaxed. As expected, none of these regions overlapped with H3K27ac peaks (Appendix A). Moreover, the majority (98%) of these regions did not occur within bivalent (i.e., co-occurring with H3K4me3 peaks in a promoter region) chromatin, which, together with the observed lack of H3K4me3 changes, suggests that there are minimal dynamic changes to bivalent chromatin in neurons with age.

## 4. Discussion

We assembled the first comprehensive maps of H3K4me3, H3K27ac, and H3K27me3 epigenetic modifications in purified neurons across an ageing time course in mice. Previous studies have used whole-brain homogenate, which profiles neurons and glia simultaneously and lacks cell specificity and clarity of insight into the epigenetic changes occurring during brain ageing. The key findings of this study were threefold; that H3K4me3 enrichment was stable, that the H3K27ac landscape exhibited a dramatic loss at enhancers in the middle-aged and old-age stages, and, similarly, that H3K27me3 was globally depleted at developmental gene regulatory regions with age.

For all the marks profiled, we found the greatest differences when comparing 3 months to 12 months or 24 months, and far fewer changes with ageing-related decline between 12 and 24 months. The current dogma of brain maturation suggests that neuronal connections are established in mice by 3 months of age [98]; however, emerging evidence shows ongoing developmental changes beyond 3 months [98,99,100]. Our data indicate a continuation of epigenetic remodelling between young-adult and mature-adult mice, with genes involved in synaptic signalling losing H3K27ac marking after 3 months of age. This is supported by reports of widespread synaptic pruning in pyramidal neurons in the cortex of mice at 3 months of age [100], and the stabilisation of adult levels of neurotransmitters and synaptic densities after 3 months of age in mice [99]. Thus, the enrichment of H3K27ac marking in young adult neurons may reflect the developmental plasticity critical for establishing gene expression profiles for stable neuronal networks in mature adult mice [99]. Considering the ages used in our data, we found that the most dramatic alterations in epigenetic marks profiled occur by middle age; however, disentangling the changes underlying both continued healthy neuronal maturation and ageing remains difficult. The loss of H3K27ac and H3K27me3 at 12 months that we observed may be reflective of both healthy maturation and more detrimental ageing; however, the epigenetic similarity of mature-adult (12 months) and old-age (24 months) neurons suggests that advanced ageing is accompanied by minimal broadly applicable changes to the epigenetic marks profiled.

H3K4me3 enrichment remained relatively static throughout neuronal ageing, with only one locus—an intronic region of *Igf2bp1*—showing the loss of H3K4me3 between 3 months and 24 months. Igf2bp1 is an RNA binding protein that binds to *β*-actin [81], and is also a known m6A reader [101]. *β*-actin has been functionally tied to synaptic function [102] and the global reconfiguration of chromatin accessibility in the brain [87]. Moreover, the RNA modification m6A has key roles in brain development and neurogenesis [101,103,104], making this gene functionally relevant to neuronal biology. Interestingly, the region of H3K4me3 loss occurs within an intron, making it unlikely to mark the active promoter for any known *Igf2bp1* transcript. Instead, we found some evidence for transcription from the opposite strand, which is potentially indicative of a novel antisense transcript. Along with our observations of changes in H3K27me3 and H3K27ac, and the consistent inclusion of synaptic signalling as a highly enriched functional term across experiments, this supports our evidence for chromatin changes occurring in advanced neuronal ageing, which may impact proper neuronal functioning.

Both H3K27ac and H3K27me3 exhibited a striking global loss in the transition between the young-adult and middle-aged stages, which was maintained through to old age. This global loss aligns with the findings of Cheng et al. [105], who reported a loss of H3K27ac in both human and mouse brain homogenates with age. We identified a decrease in H3K27ac at loci related to synaptic signalling and ion transport functions, suggesting the potential epigenetic downregulation of these processes by enhancers during neuronal ageing. This finding is consistent with evidence suggesting that synaptic function and ion transport become downregulated with age [31], and that cellular identity genes are downregulated with age [106]. Moreover, HDAC inhibitors that attenuate HDAC activity and maintain higher levels of histone acetylation have shown promise for counteracting cognitive decline, indicating the importance of maintaining higher, youthful levels of histone acetylation for neuronal function [107,108]. Interestingly, previous work has identified that H3K27ac deposition can be responsive to neuronal activity [109], which raises the question of whether lowered activity may, therefore, be a factor in age-related H3K27ac loss in neurons.

The loss of H3K27me3 that we observed indicates a global chromatin reconfiguration associated with early developmental gene programs during neuronal ageing. The link between heterochromatin loss and ageing is a long-standing observation, not only in general [110,111], but also specifically in neurons [112,113]. H3K27me3 loss in the promoters of developmental genes in particular has been recently established in both the whole brain and in the liver [114,115], and is linked to concurrent gains in DNA methylation as well as other chromatin changes, suggesting a mechanism of so-called “chromatin switching” with age. Typically, H3K27me3 and DNA methylation mark distinct loci, and DNA methylation is antagonistic to H3K27me3 deposition; however, the reverse does not appear to occur so exclusively [116,117,118,119]. In an embryonic stem cell line with reduced H3K27me3, DNA methylation was both gained and lost, with the regions that gained DNA methylation enriched for developmental terms, and largely showing no changes in gene expression [116]. This suggests that compensatory mechanisms occur to prevent the aberrant expression of these developmental genes when H3K27me3 levels are decreased, or when DNA methylation is increased. The losses of H3K27me3 that we observed were overrepresented by CpG island promoters, and whilst aged and purified neuron epigenomic data are currently limited, together with the lack of associated transcriptional changes from single-cell data, they suggest that this regulatory compensation may also occur in neurons. While the role of developmental programs in ageing is contentious, there is increasing evidence that developmental programs may play an important role in ageing alongside the accumulation of molecular damage over time [106,120,121,122,123]. Studies have shown that the epigenome is divergent throughout life [19], but cortical ageing leads to a convergence of inter-individual DNA methylation states, suggesting cell de-differentiation in the ageing human frontal cortex [124]. Transcriptomic data from the human prefrontal cortex indicate that miRNA and transcription factors regulate gene expression programs, not just during development, but across the lifespan, inextricably linking development and ageing [121,122]. This connection is exemplified by HOX genes, which are typically expressed during embryonic development and are then repressed with heterochromatin, but are shown to be expressed and functional in adult neurons [125,126]. Our observed loss of repressive H3K27me3 across several HOX gene loci in old age agrees with these previous findings, although single-cell RNA data and the lack of any enrichment of such processes in ageing neuronal cell types suggests that this may not be sufficient to drive significant changes in the gene expression of such embryonic development genes broadly [31]. Rather, the changes in H3K27me3, as well as H3K27ac, that we observed during ageing may reflect chromatin reconfiguration through a plethora of chromatin marks, and a reduction in the epigenomic plasticity as observed in cancer cells [119] and in ageing brains, which can contribute to disease risk [127].

## 5. Conclusions

Here, we present mapped H3K4me3, H3K27ac, and H3K27me3 in purified neurons across the murine lifespan. This approach, in contrast to previous “bulk”-level epigenomic investigations of brain homogenates, enabled us to dissect neuron-specific changes unobscured by the strong immune response typically mediated by glial cells. Both H3K27ac and H3K27me3 displayed a striking loss with age, in line with previous work on the whole brain. The loss of these marks, in concert with the pathways affected, suggests that genes with neuron and synapse-specific functions and those with a role in early development show a change in regulation with age. This study advances our fundamental understanding of the epigenomic biology underlying healthy neuronal ageing, a critical area of research given that ageing represents the largest risk factor for cognitive decline and the development of neurodegenerative diseases.

## Figures and Tables

**Figure 1 cells-13-01393-f001:**
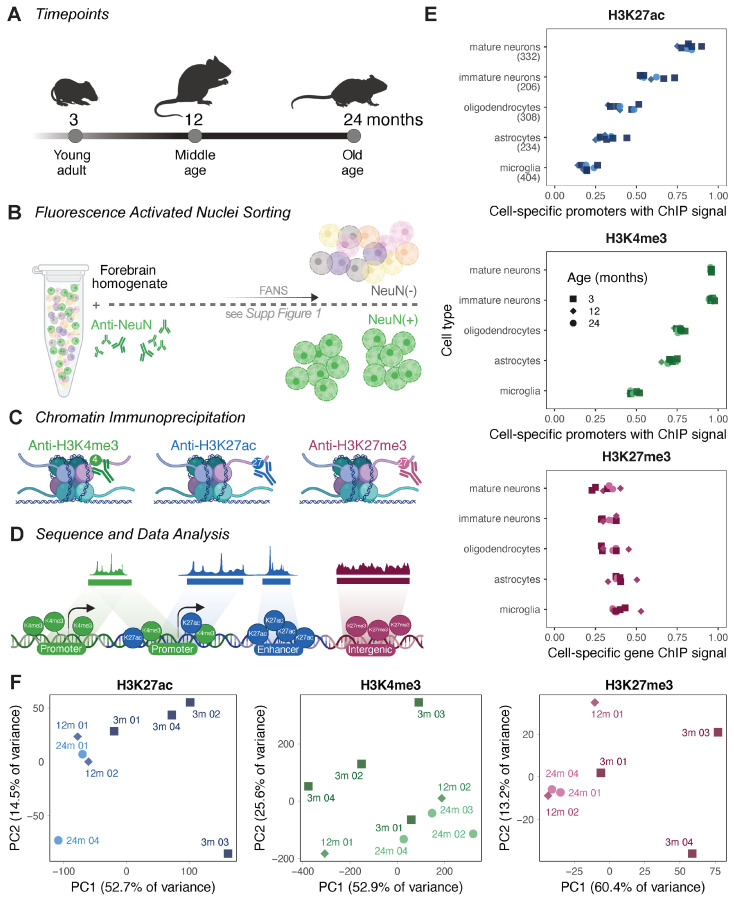
Genome-wide epigenome maps of ageing neurons. (**A**–**D**) Schematic of experimental design and analysis for this study, including (**A**) timeline of mouse lifespan and data points collected for this study, (**B**) FANS collection using NeuN+ cells, (**C**) chromatin immunoprecipitation, and (**D**) sequencing. (**E**) Enrichment of neuron-specific epigenetic signals in ageing neurons. Proportion of cell-specific gene promoters from single-cell RNA-SEQ data from Ximerakis et al. (2019) [31] with H3K4me3 and H3K27ac ChIP reads ≥ 2 RPKM, and mean H3K27me3 cell-specific gene body RPKM. Number of genes used is given in brackets after cell type. (**F**) PCA plots for genome-wide (10 kb binned) ChIP-Seq signal of H3K27ac (blue), H3K4me3 (green), and H3K27me3 (pink) in neurons.

**Figure 2 cells-13-01393-f002:**
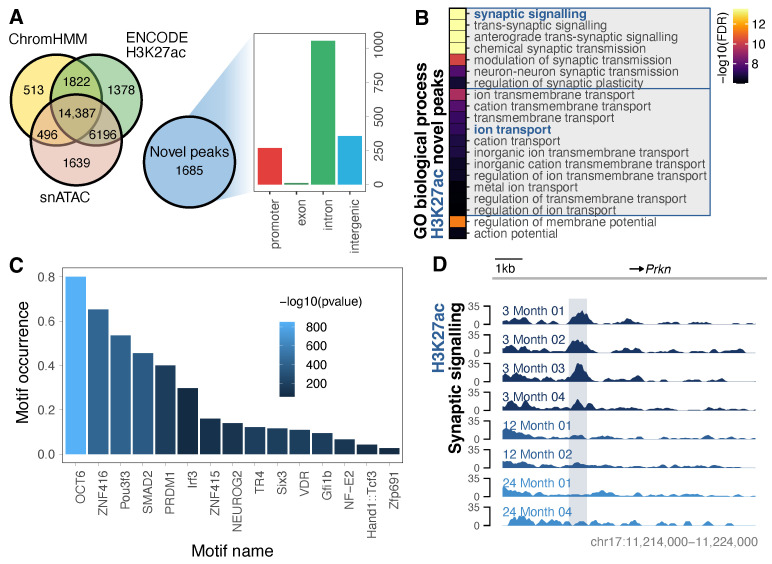
Novel H3K27ac peaks occur in synaptic-signalling-related genomic regions. (**A**) Identification of novel H3K27ac peaks that do not overlap with any of the ChromHMM enhancers, H3K27ac peaks, or snATAC peaks. Novel peaks were classed based on their genomic location, with introns being the most frequent. (**B**) Enrichment of the top 20 GO biological process terms in novel H3K27ac peaks. (**C**) HOMER de novo motif enrichment in novel H3K27ac peaks. (**D**) Neuronal H3K27ac ChIP-Seq signal in a novel peak within an intron of the synaptic signalling gene *Prkn*. Complementary H3K4me3 and H3K27me3 profiles are shown in Appendix A.

**Figure 3 cells-13-01393-f003:**
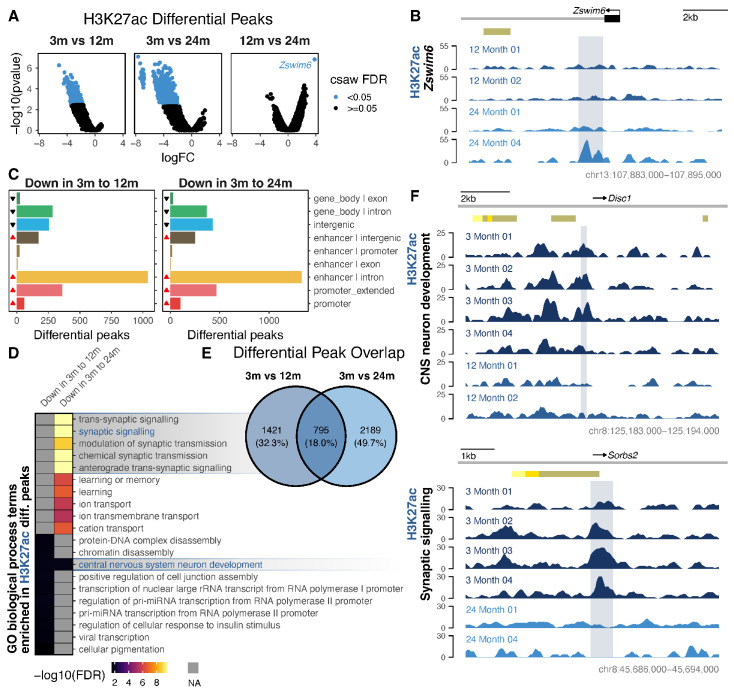
Differential H3K27ac occupancy in ageing neurons. (**A**) A csaw analysis with the glmQLFTest test identified regions of differential H3K27ac occupancy in ageing neurons from mice aged 3 months (*n* = 4), 12 months (*n* = 2), and 24 months (*n* = 2). (**B**) Region of H3K27ac gain in the first intron of *Zswim6* between 12 and 24 months. ENCODE postnatal enhancer regions are shown in yellow/gold. (**C**) Differential H3K27ac peak’s closest annotated genomic feature. Significant (*p* hypergeometric < 0.05) over- and underrepresentations are indicated by red and black triangles, respectively. (**D**) Enrichment of the top 10 GO biological process terms for each age comparison and direction of change. (**E**) Overlap of all differential peak regions in the 3-month vs. 12-month comparisons and 3-month vs. 24-month comparisons. (**F**) Example regions of H3K27ac signal, showing a loss between 3 and 12 months (*Disc1*; **top**) and between 3 and 24 months (*Sorbs2*; **bottom**). ENCODE postnatal enhancer regions are shown in yellow/gold. Complementary H3K4me3 and H3K27me3 profiles are found in Appendix A.

**Figure 4 cells-13-01393-f004:**
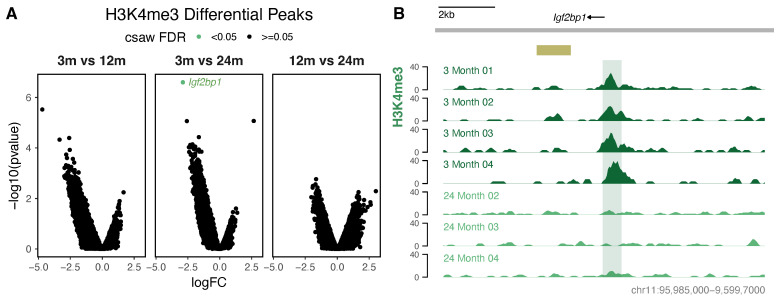
Differential H3K4me3 occupancy in ageing neurons. (**A**) A csaw analysis with the glmQLFTest test identified a single region (*Igf2bp1*) of differential H3K4me3 occupancy in ageing neurons from mice aged 3 months (*n* = 4), 12 months (*n* = 2), and 24 months (*n* = 3). (**B**) Region of H3K4me3 loss in the *Igf2bp1* intron between 3 months and 24 months. ENCODE postnatal enhancer regions are shown in yellow/gold. Complementary H3K27ac and H3K27me3 profiles are shown in Appendix A.

**Figure 5 cells-13-01393-f005:**
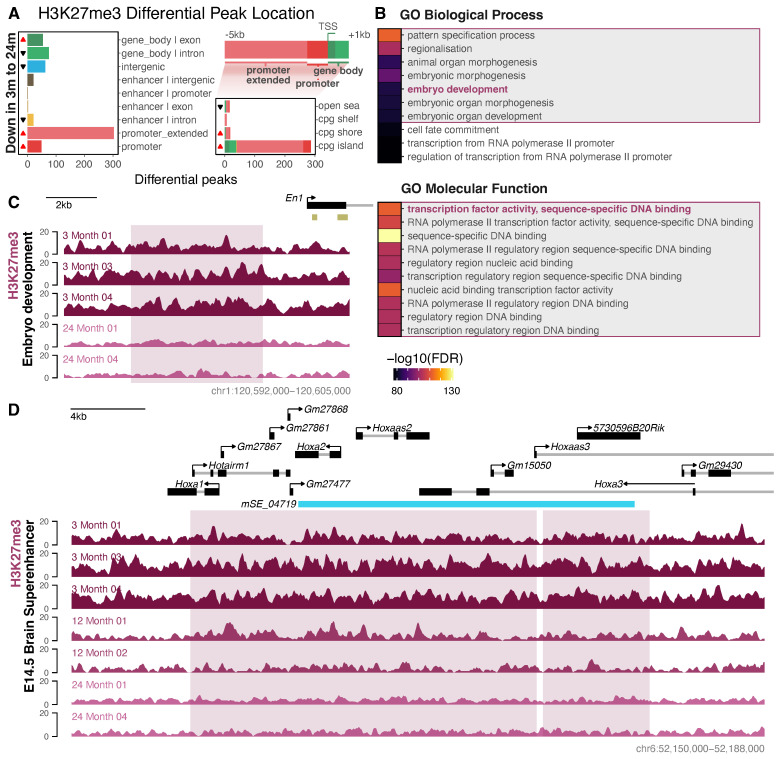
Differential H3K27me3 occupancy in ageing neurons. (**A**) Differential H3K27me3 peak locations for the 3-month (*n* = 3) to 24-month (*n* = 2) csaw analysis with the glmQLFTest. Left: closest annotated genomic feature. Right: Overlap of promoter peaks with CpG islands. Promoter overlapping peaks were defined as those overlapping the region 5 kb upstream (i.e., “promoter_extended” and “promoter”) to 1 kb downstream from the TSS (top schematic). Significant (*p* hypergeometric < 0.05) over- and underrepresentations are indicated by red and black triangles, respectively. (**B**) Enrichment of the top 10 GO biological process (top) and molecular function (bottom) terms for significant peaks in the 3-month to 24-month csaw analysis. (**C**) Example region of H3K27me signal, showing a loss between 3 and 24 months upstream of the embryonic development gene En1 promoter. ENCODE postnatal enhancer regions are shown in yellow/gold. (**D**) Differential signal of H3K27me3 at the E14.5 brain mouse superenhancer region (light blue). Regions of significant change in H3K27me3 ChIP-Seq signal are highlighted in pink. Complementary H3K27ac and H3K4me3 profiles are shown in Appendix A.

## Data Availability

The data generated in this study are available on GEO (accession GSE190102).

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
