# Peer review of "Ageing-Related Changes to H3K4me3, H3K27ac, and H3K27me3 in Purified Mouse Neurons"

_cells, 2024, doi:10.3390/cells13161393_

Round 1
Reviewer 1 Report
Comments and Suggestions for Authors
In this manuscript entitled “Ageing-Related Changes to H3K4me3, H3K27ac, and H3K27me3 in Purified Mouse Neurons”, Signal et al., used anti-NeuN(Fox-3) monoclonal antibody to mark neurons extracted from the forebrain of young adult and old age of a widely used mice model, and followed by flow cytometry sorting to obtain NeuN+ neurons (a mixture of multiple cell types) for their subsequent ChIP-seq experiments. They used the traditional ChIP-seq protocol, but now the CUT&TAG technique is more sensitive and widely used in epigenetic research for few years. CUT&TAG gives substantially better signal:noise proportion.
1. In the introduction section, what is known about histone tails marks should be mentioned and how these marks especially H3K4me3, H3K4ac, H3K36me3, H3K36ac, H3K9me3, H3K9Ac, H3K27me3 and H3K27ac generally distribute in a genome and regulate gene transcription.
2. The authors should explain properly the study design to exclude the H3K36me3, H3K9me3 analyses from the ChIP-seq?
3. Line 106-107 “Purity checks were performed and all samples were validated at >97% purity.”, but no data were shown in the supplemental info. These information is the prerequisite for proper assessing the ChIP-seq data.
4. To the sorted neurons, alternative analyses should be performed, e.g. RNA-SEQ.
5. The heatmap of ChIP-seq data should be shown for proper assessing the data
5. Flow cytometry sorting gating data should be showed for proper assessing
6. No experimental evidence to show the neurons from old mice are indeed aged compared to 3 month adult mice, proper references should be included to support this
7. in the fig2D, some ChIp-seq is deleted based on the smaple#, proper explanation should be included
8. Line120-21 “Chromatin immunoprecipitation was performed as previously described previously [25–27] with minor modiffcations.”
9. fig4A how many replicates for 3m, 12m and 24m mice and how the statistical calculation was done, in fig4B for 12m and 24m, only 2 replicates data were shown. Again, in fig4F, 12m only 2 replicates were shown.
10. The H3K27ac and H3K27me3 peaks should be showed simultaneously for a specific genomic region and compared in fig2D, fig4B, 4F, Fig5E, and also H3K4me3
Comments on the Quality of English Language
the english language is fine
Reviewer 2 Report
Comments and Suggestions for Authors
In this manuscript, Signal B and colleagues describe H3K4me3, H3K27ac, and H3K27me3 changes in mouse neurons during the aging process. The manuscript is well-written, and the datasets produced are of interest to the scientific community.
Some minor corrections are as follows:
- In the Materials and Methods section, under Sample quality control and filtering, the authors describe that 6 month-old samples did not pass quality control and were not included in further analysis. Since these samples have not been included or mentioned elsewhere, I recommend removing them from this section to avoid confusion.
- Regarding novel H3K27ac peaks (Figure 1), which datasets produced by the authors (young, 12, or 24 month-old) were used to infer this data? Additionally, why do the authors think they can annotate previously uncharacterized peaks? Perhaps different mouse strains were used in different studies? The authors should include a couple of sentences about this in the Discussion section.
- Regarding Figure 4B, the acetylation increase at Zswim6 only occurs in one 24-month sample, which undervalues the description in the text. The authors should acknowledge this variance between samples in the text.
- Regarding Figure 5, the reference in the text is too vague and should include specific references to Figures 5A, 5B, and 5C.
Reviewer 3 Report
Comments and Suggestions for Authors
The current study maps H3K4me3, H3K27ac and H3K27me3 in mouse neurons at different life stages. The data are interesting and useful to the field of epigenomic studies of ageing. However, the authors did no perform RNA-Seq analysis, which makes it difficult to interpret the potential functions of epigenomic information. Moreover, the authors did not correlate the histone modifications with functional genome elements. The detailed comments are listed below,
1. Figure 2 are discussing the newly identified H3K27ac peaks, while Figure 3 for H3K4me3, and Figure 4 for H3K27ac again. It is better to Figure 3 after Figure 4, which makes H3K27ac analysis together.
2. It is important to show the similarity between 12m and 24m groups. I suggest to redraw the figure of the PCA result and move it to the main text.
3. Although the authors show that 12m and 24m groups are similar, it is still important to identify the epigenomic difference between them, which represents the real status of ageing.
4. The analysis about H3K4me3 in Figure 3 actually does not provide much information. Could the authors present more results, such as the distribution of different peaks and the associated genes.
5. I do not understand why the authors did not perform RNA-Seq analysis. Could the authors give some explanations? Or use the online data in their study?
6. The authors simply analyzed the features for each modification, but did not consider their relationship to genome elements. Could the authors show the potential changes of different types of enhancers? Such as super enhancers and typical enhancers? Or the dynamic change of bivalent genes?
Round 2
Reviewer 1 Report
Comments and Suggestions for Authors
I think the revised manuscript has been greatly improved with the addition of more experimental control data and further focus on the novelty of experimental data. But I still feel that the authors have to do some comparisons about H3K27ac and H3K27me3 for specific genomic region and discuss; when the status of H3K27ac for a given region is depicted, the status of H3K27me3 cannot be ignored. Generally, H3K27ac and H3K27me3 two different covalent modifications should mutually-exclusive occur at the same lysine residue for a given time-point. Although the sorted neurons might be heterogeneous, I think these two marks information should be compared in ChIP-seq data, In figure2D & figure S15, figure3B,F, and explain and discuss about them in the text.
Reviewer 3 Report
Comments and Suggestions for Authors
The authors have addressed all of my concerns, and I do not have further questions.
Author Response
We thank the reviewer for their valuable comments and help in improving the manuscript.